# Examining Indigenous Identity as a Protective Factor in Mental Well-Being Research in the United States: A Scoping Review

**DOI:** 10.3390/ijerph21111404

**Published:** 2024-10-24

**Authors:** William Oyenque Carson, Caleigh Curley, Renée Goldtooth-Halwood, Deborah Jean McClelland, Stephanie Russo Carroll, Nicole P. Yuan, Scott Carvajal, Felina M. Cordova-Marks

**Affiliations:** 1Lands of the O’odham and Yaqui Peoples, Mel and Enid Zuckerman College of Public Health, University of Arizona, Tucson, AZ 85724, USA; ccurley@arizona.edu (C.C.); hozho@arizona.edu (R.G.-H.); jmcc@arizona.edu (D.J.M.); stephaniecarroll@arizona.edu (S.R.C.); nyuan@arizona.edu (N.P.Y.); carvajal@arizona.edu (S.C.); felina@arizona.edu (F.M.C.-M.); 2Lands of the O’odham and Yaqui Peoples, Native Nations Institute, Udall Center for Studies in Public Policy, University of Arizona, Tucson, AZ 85719, USA; 3Lands of the O’odham and Yaqui Peoples, Arizona Health Sciences Library, The University of Arizona, Tucson, AZ 85721, USA

**Keywords:** Indigenous identity, blood quantum, mental well-being, sovereignty, self-determination, American Indian/Alaska Native, Indigenous theory

## Abstract

Due to historical and ongoing structural racism and settler colonialism, Indigenous Peoples and communities in the United States are at a higher risk for a variety of diseases, elevated stress, and negative mental health outcomes. In addition, the United States federal government and the public encourage a view that Indigenous Peoples are primarily a racial group. Federally-, state-, and un-recognized Indigenous Peoples have a collective right to self-determination and sovereignty, and individuals of these Peoples understand this. The goals of this scoping review were to examine what research on identity and mental well-being is currently being conducted with Indigenous populations in the United States, synthesize the results, and determine if researchers are utilizing toolsets and theories that reinforce the sovereignty of Indigenous Peoples, communities, and the individual. The scoping review followed guidelines from the Joanna Briggs Institute guide for Scoping Reviews and the Preferred Reporting Items for Systematic Reviews and Meta-Analyses for Scoping Reviews (PRISMA-ScR). Four databases and over six thousand articles were searched for this review, with twenty-four that had data extracted and analyzed. Current research on the relationship between Indigenous identity and mental well-being shows mixed results. The findings of this scoping review highlight a need for Indigenous-specific tools for measuring identity in place of tools used for other ethnic and racial groups. More research must be conducted to create tools that specifically examine the phenomena of United States-based Indigenous identity.

## 1. Introduction

Indigenous Peoples in the United States are represented all throughout the forty-eight contiguous states, as well as Alaska, Hawaii, and the overseas territories. We can come from Peoples who are granted federal recognition by the United States, state-level recognition, or be unrecognized entirely with our Peoples having to petition to have our rights acknowledged. Our Peoples may legally be referred to as American Indian, Alaska Native, Native Hawaiian, Native American, Indigenous, or other terms. We may be citizens, enrolled members, descendants, disenrolled, or potentially not acknowledged. In the United States, the federal government recognizes legal standing to only a fraction of Indigenous Peoples through processes such as federal recognition of Tribal Nations and often influences Tribal Nation’s ability to determine who belongs. This can be influenced through the encouragement of Tribal Nations to adopt blood quantum or descendancy-based enrollment laws [1,2,3] and laws surrounding who can and cannot access federally sponsored projects such as healthcare services through the Indian Health Service. Across the world, colonizing nations have different views and levels of interference regarding how Indigenous Peoples are able to determine inclusion for their People, such as in Canada [4], Australia [5], or New Zealand [6]. The differences in how each set of Indigenous Peoples interacts with the imposition of their specific occupying nation warrant specific research on the issue of Indigenous identity in each country. Indigenous scholars have worked to better understand how Indigenous people’s self-identities as Indigenous people are impacted by federal government manipulation, historical trauma, and the racialization of Indigenous Nations.

One such framework is Tribal Critical Race Theory (TribalCrit). It states that Indigenous people in the US occupy a unique standing in that they possess dual classification as “both legal/political and racialized beings” [7]. Indigenous people fundamentally understand this, but it has not been fully examined as a potential modifier of an individual’s behavior. As a result, Indigenous individuals must navigate complex political and racial environments to understand their identity, while most non-Indigenous institutions/organizations and individuals in the United States cast Indigenous people as exclusively a racial group, denying their sovereignty and political status [7]. There is limited research that examines the potential role that Indigenous identity plays in mental health, and as of 16 July 2024, the search term “Tribal Critical Race Theory” in PubMed produces less than 10 results. This demonstrates there is a relative dearth of scientific literature on how Indigenous individuals come to identify themselves given the cultural and historical contexts of tribal membership. Evidence that may reduce these gaps may be addressed with appropriate Indigenous-focused theoretical frameworks such as TribalCrit.

This scoping review will examine the effects of Indigenous identity on mental well-being while also critically examining the physical locations and communities where researchers conduct studies on Indigenous identity and what measures and instruments are being utilized. This manuscript examines the current literature in health sciences, which focuses on the relationship between United States-located Indigenous identity and mental well-being through the lens of TribalCrit. Currently, there are 574 federally recognized Tribal Nations in the United States, with hundreds more state and unrecognized Tribal Nations. While there can be similarities in how Tribal Nations’ cultures and identities manifest, there continue to be differences. By attempting to understand where Indigenous identity research is being conducted, we hope to identify potential gaps. The guiding framework for this study is TribalCrit, of which tenets include that Indigenous people occupy identities that are both political, through enrollment in Tribal Nations, and racial, such as the US Census data, and that when examining culture, knowledge, and power through the lens of Indigenous they will shift, such is true with identity [7]. When utilizing TribalCrit, it becomes central to jointly consider one’s self-identity, the Tribal Nation’s enrollment laws, potential usage of blood quantum laws, and familial relations when understanding Indigenous identity. This review is limited to only examining measurements of Indigenous identity as it takes place in the United States for the following questions: 1. What is known about the relationship between Indigenous identity and mental well-being? 2. What are the measures and instruments researchers are utilizing when measuring Indigenous identity within the United States? 3. In which regions of the United States are researchers conducting their work on identity and mental well-being with Indigenous people?

## 2. Methods

This scoping review was conducted using the guidelines provided by Arksey and O’Malley as well as PRISMA-ScR [8,9]. The protocol for this review was registered with Open Science Framework’s registry for systematic and scoping review protocols [10]. The search strategy was developed in collaboration with the University of Arizona Health Sciences Library Liaison to the College of Public Health. For analysis of findings from the review process, the research team utilized TribalCrit as a guiding theory for interpreting all results [7]. This scoping review was conducted in June 2023 in PubMed, CINAHL (EBSCOhost), PsycINFO (EBSCOhost), and ProQuest Dissertations & Theses Global with the search strategy found in the Appendix A. Review inclusion criteria included the following: published within the United States, English language, full-text articles, and United States-based Indigenous Peoples, including federally recognized, state-recognized, and non-recognized Tribal Nations, as well as Alaska Native and Native Hawaiian. We limited the search to include studies with only United States-based Indigenous populations to best understand the phenomena of Indigenous identity and how it is measured and understood by researchers who work with United States-based Indigenous populations. The review did not have limitations based on the year of publication. Exclusion criteria included the following: inappropriate publication type (such as book introductions, summaries of conferences, protocol papers), non-human studies, non-English, not relevant (defined as being a human study but on an unrelated topic), Indigenous populations outside of the United States, non-Indigenous populations being used as a reference group for the research, and the research question not being addressed (Figure 1).

Title and abstract screening, full-text screening, and data extraction were conducted using DistillerSR software, Version 2.35 [11], according to PRISMA guidelines, and a final quality check was performed to ensure accuracy. To further confirm accuracy, each level of the screening process was conducted by two independent reviewers. If a conflict arose, the third member of the review team assisted in determining if the article in question would be moved to the next level of screening. Results were logged in DistillerSR after each round of screening, culminating in a table tabulated within the software, as seen in Figure 1. As a final measure to verify the accuracy of our findings, at the conclusion of the review process by the research team, we manually reviewed the list of all accepted records at the end of each level to ensure all selected materials were appropriate to advance to the next step.

At the last step of the review, data were extracted from all papers that made it through title and abstract screening, as well as full-text screening, using a data extraction tool developed by the review team (Appendix A). Two independent reviewers collected the data without consulting each other to make certain there was no bias in the results. The data extracted included specific details about the following: 1. Population demographics, 2. theoretical frameworks, 3. research questions, 4. the context, or how, when, and why the study was being conducted, 5. tools used to evaluate, and 6. key findings. The results are documented in Table 1 and further elaborated in the results of this manuscript.

## 3. Results of Review

Our search resulted in a total of 8244 records when using the search strategy in CINAHL (EBSCOhost), PsycINFO (EBSCOhost), and ProQuest Dissertations & Theses Global (Figure 1). Utilizing DistillerSR, the total number of records were screened for duplications, and 2423 were removed. The total number of records remaining after duplications was 5281.

After conducting title and abstract screening, 5738 records were excluded, resulting in 83 records for full text. Of these records in full-text screening, 59 were excluded utilizing the same criteria as the title and abstract screening phase, as seen in Figure 1. This resulted in 24 studies that advanced to data extraction. Two independent reviewers analyzed the studies based on a list of predetermined criteria using a data extraction form (Appendix A). Of these studies, 13 (54.2%) were published in the last 10 years. The other 11 (46.8%) studies were published between 1999 and 2013. Results from the review are presented in Table 1.

## 4. Results of Data Extraction Process

### 4.1. Impacts of Indigenous Identity on Mental Well-Being

All 24 studies included in this review met the criteria for inclusion as seen in the protocol [10]. These studies all examined some direct relationship between Indigenous identity and mental well-being. All but three of these studies were quantitative in nature. The three qualitative studies were also reviewed to yield complementary insights. Overall, we found that the impact of a strong Indigenous identity on well-being depended on the questions being examined, as well as the age of the populations. The study findings were categorized into three groups as follows: 1. Positive impact of identity, 2. negative impact of identity, and 3. neutral or no measured impact.

#### 4.1.1. Positive Impact of Indigenous Identity

In urban populations, research has been conducted with a variety of sub-populations and has documented a positive impact on a variety of measures of mental well-being. Three studies with urban Indigenous youth were conducted to examine moderating effects of Indigenous identity on alcohol and other drug usage [12,13,14]. In all three studies, researchers found that Indigenous identity had a positive association with decreasing drug and alcohol usage, and all recommended improving the cultural connectedness of urban youth as a potential harm reduction strategy. A study by Masotti et al. examined the effectiveness of a new scale, the Cultural connectedness scale-California (CCS-CA), and found that Indigenous culture was an important determinant of health and linked to well-being [15,16].

Four studies were conducted specifically with Alaska Native populations that examined acculturation, sense of community, and coping strategies [17,18,19]. In the two Wolsko et al. studies, the researchers identified and confirmed that Yup’ik communities associate Indigenous ways of knowing as a protective factor in relation to mental well-being and historical loss. A more recent study by Rivkin et al. saw the development of community-driven resources and interventions that document culture and Indigenous identity as a sense of strength for Yup’ik communities. Finally, a study conducted by Buckingham et al. with Alaska Native undergraduate students at a college in Alaska saw that elder-led programs to improve students’ ties to their communities and enhance Indigenous identity had positive results and improved overall emotional health of those who participated [20].

Multiple studies were conducted with Cherokee Tribal Nations, including the Hoffman et al. study and Lewis et al. studies with the Eastern Band of Cherokee Indians [21,22] and the Lowe study with the Cherokee Nation in Oklahoma [23]. In the Hoffman and Lowe studies, the relationship between Cherokee identity and mental well-being was measured. In the Lowe study, the Cherokee Self-Reliance Questionnaire [23] was utilized, while the Hoffman study used a modified version of the Multidimensional Inventory of Black Identity [21]. In one study, the main outcome was perceived stress [23]. Another study measured self-esteem and overall mental health as the main outcomes [21]. In both cases, a stronger sense of Cherokee identity was positively associated with lower levels of stress/higher self-esteem [24,25]. Lewis et al. assessed the effectiveness of Cherokee culture-based programs on health and well-being; researchers found that participants in the program had improved exercise and diet behaviors, implying that Indigenous culture is an effective preventative measure [22].

Finally, a study with Navajo youth enrolled at Navajo Nation schools found that higher levels of Navajo identity had a positive effect on reducing depressive symptoms, although more research is needed to fully understand this relationship [26].

#### 4.1.2. Negative Impact of Indigenous Identity

In 2003, research to investigate the relationship between self-reported anxiety, stressful events, and cultural identification among Northern Plains adult populations commenced [24]. In the study, researchers found that cultural identification did not buffer the relationship between stressful life events and anxiety [24]. In addition, in a study conducted with Alaska Native populations examining the effects of discrimination on acculturative stress, it was noted that increased levels of discrimination and acculturative stress were seen with increased Indigenous identity [27].

In a study by Ehlers et al., their goal was to evaluate if thoughts of historical loss and associated symptoms are influenced by variables such as cultural identification and percentage of Native American ancestry, among other items [25]. In this reservation-based study, it was found that historical loss is associated with stronger cultural identification, with one-fourth of participants thinking daily about the loss of culture, language, and respect for elders [25].

In 2019, researchers at a southwestern university conducted with Indigenous (most identifying as Navajo) undergraduate students found that Indigenous identity and belonging are all negatively related to academic stress, with students believing their values did not fit in with the rest of the university [28]. These students were recruited through student support services and screened through self-identification as Indigenous. The authors noted that the promotion of self-identity and a sense of belonging may be beneficial [28].

#### 4.1.3. Neutral or No Reported Impact of Indigenous Identity

The earliest studies in this section were from 1999 to 2000, in university settings in the Navajo Nation and an unnamed college in Kansas. The authors of these articles found that there was no significant relationship between Indigenous identity and cultural/social anxieties [29,30]. Research conducted with boarding schools in the southwest investigated the relationship between stressful life events and Indigenous identity [31]. The findings were mixed. Cultural identity was not directly associated with a protective factor, but social support, protective family ties, and peer influences had some positive impacts. The researchers recommended further investigation into these specific areas [31].

In one study conducted at a large southwestern university, researchers examined how Indigenous students’ identities may impact their experiences and successful matriculation as college students [32]. The results of this study found that there was little change in Indigenous identity and that those with a stronger sense of Indigenous identity had higher levels of stress [32]. Reynolds et al. intended to validate an instrument to incorporate Indigenous cultural values as they relate to mental health issues among Dakota, Nakota, and Lakota university students but were unable to test their hypothesis due to small sample sizes and issues with the implementation of the study [33]. Finally, in dissertation work exploring the moderating effect of Native Hawaiian identity, it was documented that life events and Native Hawaiian identity-related stressors predicted increased levels of depression and anxiety while not finding Native Hawaiian identity to moderate these phenomena [34].

### 4.2. Geographical Location and Populations: Where and with Whom Are These Studies Being Conducted?

One of the main goals of this scoping review was to understand in which regions and with which specific Indigenous communities are researchers conducting studies on identity and mental well-being. We split the ‘where’ into three unique categories: 1. Indigenous lands, such as reservations, within Alaska Native villages and Native Hawaiian Communities; 2. urban areas; 3. dorms, colleges, and universities. The numbers and representation for this section will be different from that of Section 4.1 as the prior section was divided by the age of participants while this section is divided by region and location.

#### 4.2.1. Reservations, Alaskan Native Villages, and Native Hawaiian Communities

Eleven studies in this review were reservation-based or within Alaskan Native villages. Four studies conducted with Indigenous adolescents examining identity’s influence on self-esteem and depression were conducted with youth from the Eastern Band of Cherokee Indians in North Carolina, Cherokee Nation in Oklahoma, as well as Navajo youth in reservation-based schools [21,22,23,26]. In California, researchers conducted a study with Indigenous people above a certain blood quantum threshold who lived in one of eight contiguous reservations [25]. Three studies taking place in Alaska were conducted exclusively within Yup’ik communities and only with self-identified residents of Yup’ik villages [17,18,19]. Another examined the impact of perceived discrimination on the stress of Indigenous Alaskans was conducted in rural villages, which included Aleut Yup’ik, Inupiaq, Athabascan, Tlingit, and Haida people [27]. McCubbin’s research was conducted with Native Hawaiian youth from a singular high school in Hawaii [34]. However, not all research published listed the Tribes or locations in which the studies were conducted. One article conducted with a rural community in Nebraska specified that 82% of the population was from the same Tribal Nation [24].

#### 4.2.2. Urban Areas

Five studies were conducted with urban Indigenous populations, with every study taking place in California. Three of these studies described urban Indigenous youth’s identities, and other factors, with more than sixty total tribes represented [12,13,14]. To be eligible for these studies, individuals had to be between the ages of 14 and 18 and identify as American Indian or Alaska Native. A fourth study in California was focused within the San Francisco Bay Area, where research was with adult Indigenous people from 107 different tribal affiliations ranging from ages 18 to 79 [16]. The fifth study in the state looked at Indigenous culture as it manifests within urban dwelling Indigenous adults, ages ranging from 18 to 84, within San Francisco, Santa Rosa, Fresno, and outlying areas [15].

#### 4.2.3. Dorms, Colleges, and Universities

Eight studies in this review examined Indigenous populations on college/university/high school campuses and dormitories. One study conducted in the southwest was with youth, ages 15–24, residing in two boarding school dormitories for high schoolers located off-reservation [31]. Per the confidentiality agreements made with Tribal Nations collaborating in this research, the exact location of the schools and Tribal affiliations of the youth were not disclosed [31]. Studies focused on college and university settings have taken place all over the country. Research has been conducted within universities in the state of Arizona [28], Kansas [30], Oklahoma [32], Alaska [20] as well as universities within Tribal Nations such as the Navajo Nation [29]. Another two studies were conducted across multiple colleges, one being open to all Indigenous-identifying students [35] with the other only being conducted with Dakota/Nakota/Lakota identifying students [33].

### 4.3. Measuring Identity: Tools of the Trade

The final aim of this scoping review was to examine the types of tools utilized by researchers when measuring Indigenous identity in the United States. Twenty-one out of the twenty-four studies in this review were quantitative in nature and utilized identity scales. Three studies utilized qualitative methods, and while important to understanding identity and well-being, did not have a scale or measure to report on for this analysis but instead reported based on results from the thematic analysis [13,17,18]. A wide variety of tools were utilized throughout these studies; researchers implemented twelve separate tools. Seven of these tools were adapted for specific tribes/Indigenous communities through swapping out terminology. Five of these tools were new tools that were created specifically for Indigenous populations.

The Multigroup Ethnic Identity Measure (MEIM) and the Multigroup Ethnic Identity Measure-Revised (MEIM-R) are brief surveys used to assess the individual’s relationship with their self-identified ethnic groups [36]. Several researchers utilized this survey for a variety of different populations, including urban youth [12,14,34], university students [20,29,32,35], and reservation-based individuals [22]. Another scale used in two studies was the Orthogonal Cultural Identification Scale (OCIS) [25,31]. The OCIS explores the strength of identification with each culture that a participant has separately [31].

In other cases, researchers modified surveys implemented for non-Indigenous populations and made them fit the goals of the research. In a study measuring the identification of Alaska Natives and perceived discrimination and acculturative stress, the research team adopted Branscombe’s gender identity scale by replacing “gender group” with “Alaska Native” [27]. Two studies adapted the Multidimensional Inventory of Black Identity (MIBI-T) [21,28]. As the original scales were used for Black populations, researchers changed racial labels from “Black” to an Indigenous term that fit their studies, such as “Cherokee” or “Native American” [21,28].

Finally, many studies developed novel scales for use with Indigenous populations, with six scales developed before 2010. As early as 2000, Indigenous identity scales were being utilized, such as the Native American Cultural Involvement and Detachment Anxiety Questionnaire (CIDAQ) that measured cultural involvement, social involvement with tribes, interaction with the dominant culture, knowledge of Indigenous culture, among other items [30]. Three Tribal Nation specific scales were developed between 2003 and 2006, Northern Plains Bicultural Inventory [24], Navajo Cultural Identity Measure (NCIM) [26], and the Cherokee Self-Reliance Questionnaire [23]. Each team created a tool that would measure cultural identity and connectedness with the nation involved in the study. In a study with Dakota/Nakota/Lakota people, the researchers piloted the Native American Cultural Values and Beliefs Survey (NACVBS) [33]. The NACVBS is a survey that measures group identity, individual identity, language, values, and beliefs [33]. A 2007 study out of Alaska developed scales to measure Yup’ik and Kass’aq (white) identification to examine acculturation and enculturation [19]. More recently, in multiple studies in California, researchers utilized the Cultural Connectedness Scale-California (CCS-CA) scale, which measures Native culture and cultural connectedness as well as mental and physical health factors [15,16].

### 4.4. Explicit Explanation of Theoretical Frameworks Used

An additional theme identified in this review is the identification of explicit explanations of theoretical frameworks utilized by researchers in this scoping review. When we examined which articles had utilized TribalCrit to guide their work, only one article of a potential 17 had mentioned the theory [32]. The four articles did not mention any type of theoretical framework guiding their research.

The most commonly used theoretical framework in the studies reviewed was Orthogonal Cultural Identification Theory (OCIT), which was used to study drug usage in Indigenous and Mexican American youth and posits that cultural identities exist independently of each other and are most often hypothesized that identification with any culture may provide a buffer against negative indicators of well-being [37] (in contrast to a state of marginalization from multiple cultures, assumed to place persons at risk). This may be considered an effective theory for making inferences around broad ethnic groups, but OCIT does not address the political nature of Indigenous identity, such as themes on Tribal enrollment, blood quantum, or other contemporary issues among Indigenous people.

In seven studies there were Indigenous methodologies utilized, such as Indigenist Stress-Coping, TribalCrit, and framing work around historical trauma [12,22,26,32,35], with another two articles with less structured theory centering Indigenous knowledge and ways of knowing [14,33]. Other studies involved utilized theories such as community-based Participatory Research, grounded theory, biculturalism, and social identity theory, amongst others. While these approaches may have potential, they also may have limitations without explicit relevance to Indigenous people’s experiences, historical context, political factors, and specific Tribe’s culture, which is vital for understanding Indigenous identity.

## 5. Discussion

This scoping review was conducted to better understand how Indigenous identity has been measured in research in the United States and its relationship with mental well-being or mental health indicators, such as stress, depression, and anxiety, among others. However, after a review of the tools utilized and the theoretical frameworks involved, there are important opportunities for future work to address. What we see in this review is a mixture of results that show Indigenous identity as positive or negative depending on the context and questions being asked. What we also observed in this review is that depending on the population, the scales used, and the research questions asked, the impact of Indigenous identity on mental well-being changes greatly. In a wide range of studies here, we saw findings that showed a negative correlation between Indigenous identity and measurable well-being outcomes. A strong Indigenous identity may lead to lower perceived health status [24], elevated levels of stress [28,34], and increased feelings of historical loss [35]. Participants saw improved mental health when they had a stronger Indigenous identity in studies examining how to foster Indigenous identity through interventions in youth populations [16,17,20,21,22]. Many of the studies utilized the same tools; we observed that results changed based on how identity was measured and for what purpose.

Contrary to an initial analysis, these results may not contradict each other. Indigenous identity, the embrace of it, means also embracing the challenges of understanding our past, of which historical trauma is highly researched [38,39,40,41,42]. Indigenous identity may offer a protective nature against mental well-being, but also, by actively being involved with your identity and community, an Indigenous person may also be more susceptible to the challenges facing the tribe and experience contemporary structural racism. Theresa O’Nell, an ally scholar (non-Indigenous researcher) who did work with the Flathead Indian Tribe, developed the following model, which may help in explaining the simultaneous benefits and challenges of a strong Indigenous identity. The O’Nell model of the Empty Center describes Indigenous identity as a constantly shifting ideal of what modern Indigenous people strive to be, with the younger generations viewing the older generations as being knowledge keepers and “[M]ore Indigenous” [43]. For the work of O’Nell, this ideal for the Flathead people was the practice of traditional customs, consuming traditional foods, and speaking their language [43]. As noted in her work and others, Indigenous people would continue to strive to embrace their cultures and traditions, but with each passing generation and the slow encroachment of Western culture, the Indigenous identity was both a source of strength and struggle. This was documented in 1999, and additional work in other fields of study continues to acknowledge the challenge of colonization on Indigenous identity, but the field of health sciences remains behind.

This review compiled and summarized all available literature to better understand how the scientific community measures Indigenous identity. When examining the results through the lens of TribalCrit, the review highlights major gaps in tools that understand Indigenous identity and from Indigenous, tribally-centered, lived experience. In the early 2000s, several Indigenous-led models were developed and used, but this review did not show sustained utilization and continuous development [23,24,26]. Survey length may play a role in decreased utilization of instruments. The NCIM has 152 questions and is specifically aimed at Navajo people, so the specificity and enormous size of this survey may be reasons for the lack of subsequent use [26]. What we continue to see when examining the nature of these scales is that whether they are Indigenous framed or adopted from a more generalizable tool, limitations remain. These scales and tools most often ask broad questions about an individual’s identity and cultural attachment but fail to ask nuanced questions related to the effects of history, colonization, and forced adoption of policies such as blood quantum. TribalCrit is a decolonial theory in which Dr. Bryan Brayboy lays out how Indigenous identity has been attempted to be regulated by the federal government and that US policy toward Indigenous people is grounded in imperialism, white supremacy, and material gain [7]. While the studies utilize promising tools, it is not evident that they capture the challenges Indigenous people have every day in fighting for their identities to be acknowledged as valid and recognized as holding political power, nor the strengths and beauty in Indigenous cultures. By adopting scales used for other broad ethnic and racial group categorization, as well as typically failing in developing identity scales situated in the political identity of Indigenous communities, it could be asked whether much of the existing tools and practices in this area of research are truly contributing to legitimizing Indigenous Nations as sovereigns and adequately, considering Tribal culture and historical context.

The research team observed a curious occurrence in how scales were developed when we framed the work utilizing TribalCrit. The landmark paper by Dr. Brayboy, which established the theory of TribalCrit, was published in 2005, which was the same time period when the Tribal Nation-specific scales were being developed by various researchers. These studies, published between 2003 and 2006 [18,23,24,26,33], would have all been well underway when TribalCrit was first published, meaning that they would not be able to incorporate this theory’s understanding of “dual identity” unless separately determined. This is unfortunate for this study as the only articles after TribalCrit all utilized scales from the general population and modified for their work with Indigenous people, as seen in Table 1. More interesting is that TribalCrit was not mentioned as a theoretical framework for any study until 2023 [32].

Finally, when examining where these studies are conducted, we noticed that in one-third of the articles, tribal affiliation was not specified. We understand that this is a potentially complicated issue when working with Tribal Nations, as they may choose to be de-identified. Important to note that data about Indigenous people, even outside of Tribal Nation boundaries, is still data about Indigenous people, and therefore, researchers should respect Indigenous Data Governance guidelines such as the CARE and FAIR Principles [44]. When potentially reporting out tribal affiliation data, the necessary Tribal Nations should be communicated with. There was a large variety of Indigenous people and Nations represented in this scoping review, representing federal, state, and unrecognized Indigenous people, and in most cases, the Tribal Nation affiliations were stated directly. Sixteen of twenty-four studies listed the Tribal Nation affiliations of the participants in this study. However, four of these did not, rather stating in their articles that these findings are not listed, not available, not disclosed, or not made public [13,15,25,31]. In addition, four studies did not mention any tribal affiliations by name but instead stated the total number of the tribal affiliations of people who participated, and in three instances, the only notes the authors gave in their findings were about tribal affiliation where a number of tribes represented [12,14,15,35]. In a singular instance, we noted that the authors did not list out all participant tribal affiliations but did specify the Tribal Nations where the majority of participants came from [32].

We recommend future research include Indigenous-centered theory and lived experience in research focused on Indigenous people, especially in sensitive areas such as identity. Given the gaps and limitations identified in this review, we suggest more inroads in understanding and promoting Indigenous persons’ well-being with a more focused approach rather than one with models and measures adapted from other populations.

## 6. Challenges and Limitations

One of the main challenges of this review is the potential to have missed relevant studies. This is due to the selection of databases to search, given that the ones selected primarily address the health and behavioral sciences fields. It may be advantageous to conduct a vigorous hand search of American Indian Studies journals and databases in future studies of this nature. We may have been able to counteract this due to our work with the University of Arizona Health Sciences Library, as our initial screening went from just over one thousand articles to over seven thousand with their help. A second potential limitation of this study was our inclusion and exclusion criteria. We are aware that research on Indigenous identity is being conducted in other parts of the world, and due to our choice to limit research to just the United States, our findings may have been less conclusive than those of a global review. However, this is justified given our interest in blood quantum laws and the United States federal government’s involvement in enrollment processes and legal recognition of Tribal Nations. While all Indigenous communities have faced forms of colonialism, it is important to emphasize each colonial nation’s unique treatment of Indigenous people.

## 7. Future Directions

The findings in this review demonstrate that more research is needed to understand Indigenous identity as it relates to mental well-being. We note that in many studies, there is a lack of mention of Indigenous-led theory, which is of utmost importance when discussing the complex field of Indigenous identity. TribalCrit should be more closely examined and implemented by researchers, and to properly implement it, our Indigenous people and scholars need to be involved in future work. It is critical for research around something as sensitive and timely as an Indigenous identity to include meaningful authorship from Indigenous researchers, tribal leaders, and community members with relevant lived experience.

## 8. Conclusions

We identified gaps in the current research examining identity in United States-based Indigenous populations and mental well-being. There are limitations in a number of studies and the understanding of Indigenous identity with contributions from Indigenous Peoples within the United States. Researchers have examined the role of Indigenous identity in a variety of settings with populations across the country. However, most often, the tools used have limitations in that they do not express important aspects of Indigenous identity. Indigenous identity research may benefit from the frameworks such as TribalCrit included in this review. What we observed here is that the theoretical frameworks used to examine Indigenous identity are often unspecified. This manuscript highlights a need for researchers to develop Indigenous-specific tools that include both the racial and political aspects of US-based Indigenous peoples’ identities and are grounded in Indigenous people’s lived experiences.

## Figures and Tables

**Figure 1 ijerph-21-01404-f001:**
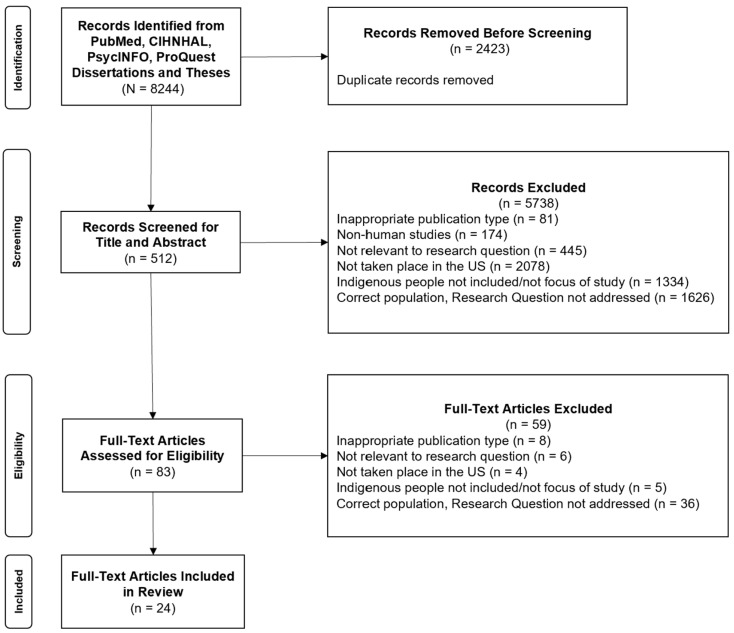
Results from scoping review screening process.

**Table 1 ijerph-21-01404-t001:** Results of the scoping review data extraction process.

Title	Year	What Is the Objective of the Paper?	Concept(s) Being Studied	Inclusion/ Exclusion Criteria	Population	What Is the Tribe(s) or Urban Areas Involved	Identity Scale(s) Used	How Are Indigenous People Defined in the Study?	What Theoretical Frameworks Are Utilized for the Study?	Does the Paper Mention if Any Authors Are Indigenous?
Culturally related anxiety and ethnic identity in Navajo college students	1999	To explore the relationship between ethnic identity and culturally related anxiety	Identity; Personal Health; Well-Being	Self-Identity Of Navajo University Students	Navajo University Students	Navajo Nation	Native American Cultural Involvement and Detachment Anxiety Questionnaire (CIDAQ); Multigroup Ethnic Identity Measure	Self-Identity	Orthogonal Cultural Identification Theory	No
Assessment of culturally related anxiety in American Indians and Alaska Natives	2000	To develop and provide an initial test of the Native American Cultural Involvement and Detachment Anxiety Questionnaire (CIDAQ) in a sample of American Indian and Alaska Native college students	Perceived Discrimination; Acculturative, And Physical Stress	Native American and Alaska Native College Students in Kansas	AI/AN University Students	Haskell University Students	Native American Cultural Involvement and Detachment Anxiety Questionnaire (CIDAQ); Cultural Identification Scale	Self-Identity	Orthogonal Cultural Identification Theory	No
Anxiety, stress, and health in northern plains Native Americans	2003	To investigate the relationship between self-reported anxiety, stressful events, health, and cultural identification among Native Americans from a rural community in northeastern Nebraska	Cultural Identity’s Association with Alcohol and Other Drug Use	Enrollment Or Reported Family Lineage and Community Recognition	Northern Plains Native Americans	Various Tribes: 85% Of Participants Indicated They Were from The Same Tribe	Northern Plains Bicultural Inventory	Reported Enrollment in A Federally Recognized Tribe or Reported Family Lineage and Community Recognition.	Orthogonal Cultural Identification Theory; Orthogonal Theory of Biculturalism	Yes
Resilience among native Hawaiian adolescents: ethnic identity, psychological distress, and well-being	2003	To examine the effects of stressful life events and cultural stressors on psychological functioning To examine effects of ethnic identity as a protective factor on psychological functioning Examine if ethnic identity moderates’ effect of stressors on psychological functioning	Identity; Tradition; Spirituality; Wellbeing; And Mental Health	Native Hawaiian Students from Singular High School, Ages 14–18	Native Hawaiians	Native Hawaiians	Multigroup Ethnic Identity Measure (MEIM)	Self-Identity	Model Of Resilience	No
Cultural identity, explanatory style, and depression in Navajo adolescents	2004	To understand the relationship between Navajo cultural identity and depression and its risk factors	Cultural Connectedness; Physical Health; And Mental Health	Navajo Adolescent Students Enrolled in Navajo Reservation Schools	Navajo Adolescents	Navajo	Navajo Cultural Identity Measure (NCIM)	Students Enrolled at Navajo Schools and Self-Identify	Indigenous Stress and Coping Model	No
Initial development of a Cultural Values and Beliefs Scale among Dakota/Nakota/Lakota people: a pilot study	2006	Validate a tool designed to Indigenous incorporate cultural values into a mental health instrument	Mental Health; Stress and Coping; Historical Loss; Ethnic Identity; Well-Being	Self-Identification Of Current and Former Dakota, Nakota, And Lakota University Students	Dakota/Nakota/Lakota University Students	Dakota/Nakota/Lakota in Midwestern University Setting	Native American Cultural Values and Beliefs Survey (NACVBS)	Self-Identity	Wellness Grounded in Spirituality, Values, And Beliefs	No
Teen Intervention Project–Cherokee	2006	To advance knowledge for practice concerning alcohol abuse intervention among a Native American adolescent population	Cultural Identity and Beliefs/Level of Distress	Cherokee Adolescents Who Were Referred from Substance Abuse Counseling at School	Cherokee Adolescents	Cherokee Nation	Cherokee Self-Reliance Questionnaire	Enrollment	Social Learning Theory; Problem Behavior Theory	Yes
Stress, coping, and well-being among the Yup’ik of the Yukon-Kuskokwim Delta: the role of enculturation and acculturation	2007	To report on the relationships between cultural identity and stress, coping, and psychological well-being in Yup’ik communities	Identity; Stress, Coping; Psychological Well-Being	Self-Identity	Yup’ik People in Six Rural Villages in Alaska	Yup’ik People in Alaska	Measures Of Cultural Identification Consisted of Two Separate Items. One Item Assessed the Level of Kass’aq (White) Identification or Acculturation. A Second Item Assessed Level of Yup’ik Identification, Or Enculturation	Self-Identity, From Village	Orthogonal Cultural Identification Theory	Yes
Conceptions of Wellness among the Yup’ik of the Yukon–Kuskokwim Delta: The Vitality of Social and Natural Connection	2007	To understand the plausible health benefits of enculturation, including the role of cultivating harmonious relationships, core values, and self-defining features of the traditional Yup’ik worldview	Identity; Mental Health; Well-Being;	Yup’ik Adults	Yup’ik Indigenous People	Yup’ik Indigenous Communities	Qualitative Study	Self-Identified Yup’ik People Residing in Villages	Grounded Theory	No
Culture and Context: Buffering the Relationship Between Stressful Life Events and Risky Behaviors in American Indian Youth	2011	The Sacred Mountain Youth Project was conducted to investigate risk and protective factors related to alcohol and drug use among American Indian youth	Identity; Resilience; Stress; Alcohol and Drug Use	Must Be American Indian and In One Of Two Eligible Dormitories Between the Ages Of 15 And 24	Indigenous Youth	Not Disclosed	Orthogonal Cultural Identification Scale	Self-Identity	Orthogonal Cultural Identification Theory	Yes
Measuring historical trauma in an American Indian community sample: contributions of substance dependence, affective disorder, conduct disorder, and PTSD	2013	To evaluate the extent to which the frequency of thoughts of historical loss and associated symptoms are influenced by current traumatic events, post-traumatic stress disorder (PTSD), cultural identification, percent Native American Heritage, substance dependence, affective/anxiety disorders, and conduct disorder/antisocial personality disorder (ASPD)	Trauma; PTSD; Cultural Identity; Substance Dependence	At Least 1/16th Native American Heritage (NAH), Be Between the Ages Of 18 And 70 Years, And Be Mobile Enough to Be Transported from His or Her Home to The Scripps Research Institute (TSRI)	Reservation Based Indigenous People	Not Made Public	Orthogonal Cultural Identification Scale (OCIS)	Blood Quantum Based	None Listed	No
Cultural Identity among Urban American Indian/Native Alaskan Youth: Implications for Alcohol and Drug Use	2016	To develop culturally relevant and developmentally appropriate alcohol and other drug use interventions for urban AI/AN youth	Alcohol And Substance Use; Risky Behavior; American Indian Youth; Cultural Identity; Depressed Mood; Risk Factors; Protective Factors	Does Not Explicitly Say but Participants Are AI/AN From Northern or Southern California and Of All Different Ages.	Urban AI/AN Youth, Parents, Providers in California	Urban Communities in Northern and Southern California	Qualitative Study	Self-Identification And Recruitment Through Third Party Health Services	Historical Trauma; Community Based Participatory Research	No
Cultural values, coping, and hope in Yup’ik communities facing rapid cultural change	2018	To build knowledge of culturally based strategies, values, resources, and protective factors that facilitate adaptation to rapid cultural change and that could be used in community-driven interventions to promote wellness in rural Alaska Native communities	Culture; Coping; Intergenerational Knowledge; Historical Trauma; Resilience Factors	Residents Of Yup’ik Villages	Yup’ik Communities in Alaska	Yup’ik Tribe	Qualitative Study	Self-Identity Of Indigenous People Within Communities	Grounded Theory	Yes
Academic stress of Native American undergraduates: The role of ethnic identity, cultural congruity, and self-beliefs	2019	To explore the relation of self-beliefs, ethnic identity, and cultural congruity with academic stress amongst Indigenous university students	Cultural Identity; Sense of Community; Emotional/Behavioral Health	Self-Identification, Recruitment Through Student Support Services	Indigenous Undergraduate University Students	University Setting	Modified version of Multidimensional Inventory of Black Identity (MIBI-T)	Self-Identity	Theory Of Academic Persistence	Yes
The Culture is Prevention Project: Measuring Culture as a Social Determinant of Mental Health for Native/Indigenous Peoples	2020	Will the incorporation of Indigenous culture affect health outcomes for Indigenous people	Ethnic Identity; College Outcomes; Academic Success	Self-Identification	San Francisco Bay Area Indigenous Community Members	Urban Indigenous Population in The San Francisco Bay Area And the Surrounding Area	Cultural Connectedness Scale-California (CCS-CA)	Self-Identity	Community-Based Participatory Research Approach; Strength-Based Approach	Yes
Indigenous Alaskan and mainstream identification explain the link between perceived discrimination and acculturative stress	2021	To examine the effect of discrimination on acculturative and physical stress	Ethnic-Racial Identity; Gender Identity; Adolescence; American Indians; Psychosocial Adjustment	Self-Identification	Alaska Native Identifying People	Aleut, Alaska Native, Eskimo, Yup’ik, Inupiaq, Athabascan, Tlingit and Haida	Branscombe’s Gender Identity Scale, Replacing “Gender Group” With “Alaska Native”	Self-Identity	Rejection-Identification Hypothesis; Acculturation Hypothesis	No
Identifying as American Indian/Alaska Native in Urban Areas: Implications for Adolescent Behavioral Health and Well-Being	2021	To examine Indigenous identity’s association with behavioral health and well-being	Ethnic Identity; Academic Stress; Cultural Congruity; Self-belief	Adolescents Had to Be 14 To 18 Years Old (Inclusive) And Either Verbally Identify as AI/AN Or Be Identified As AI/AN By a Family Member.	Urban Indigenous Adolescents	Urban Areas Across Central, Southern, And Northern California	Multigroup Ethnic Identity Measure (MEIM)	Self-Identity	Theoretical Conceptualization of Behavioral Health and Well-Being as It Relates To AI/AN Racial-Ethnic and Cultural Identity	No
Unveiling an ‘invisible population’: health, substance use, sexual behavior, culture, and discrimination among urban American Indian/Alaska Native adolescents in California	2021	To address gaps in urban Indigenous communities by sharing qualitative research with urban Indigenous communities	Cultural Identity; Stress; Discrimination; Urban/Reservation	To Be Eligible for The Project, Adolescents Had to Be 14–18 Years Old (Inclusive) And Either Verbally Self-Identify as AI/AN Or Be Identified As AI/AN By a Parent/Guardian or Community Elder. Eligible Adolescents Were Scheduled to Complete a Baseline Survey at A Time and Place That Was Convenient to Them.	Urban Indigenous Adolescents	Northern, Central, And Southern California Urban Areas	Multigroup Ethnic Identity Measure (MEIM)	Self-Identification Or Identification by Elder or Parent	None Specified	No
Ethnic-racial identity, gender identity, and well-being in Cherokee early adolescents	2021	Examine Cherokee adolescents’ ethnic identities and how they relate to self-esteem and well-being	Stress; Cultural Identity; Anxiety	Citizens Of Eastern Band of Cherokee Indians	Cherokee Adolescents	Eastern Band of Cherokee Indians	Modified version of Multidimensional Inventory of Black Identity (MIBI-T)	Enrollment	Self-Categorization Theory; Social Identity Theories	No
The Health Effects of a Cherokee Grounded Culture and Leadership	2022	To assess the effectiveness of Cherokee culture-based program on changes in health and well-being	Cultural Identity; Anxiety; Stress; Involvement in Western Culture	Participants Selected from Their Respective Tribes	Cherokee Nation Youth	Cherokee Nation; Eastern Band of Cherokee Indians	Multigroup Ethnic Identity Measure	Enrolled Members Selected by Tribal Authorities	Indigenous/CBPR Outcomes Evaluation	Yes
The Culture is Prevention Project: measuring cultural connectedness and providing evidence that culture is a social determinant of health for Native Americans	2023	To report on Culture is Prevention and the assessment of cultural connectedness, physical health, and mental health	Anxiety; Ethnic Identity; Culture	Self-Identification	Urban Dwelling Native Americans	San Francisco, Santa Rosa, Fresno	Cultural Connectedness Scale-California (CCS-CA)	Self-Identity	None Specified	Yes
The Impact of Historical Loss on Native American College Students’ Mental Health: The Protective Role of Ethnic Identity	2023	To examine the theorized pathways among historical loss, well-being, psychological distress, and the proposed cultural buffer of ethnic identity in the indigenist stress-coping model (ISCM)	Well-being; Ethnic Identity; And Psychological Distress	Not Explicitly Stated	Indigenous University Students	University Setting	Multigroup Ethnic Identity Measure-Revised (MEIM-R)	Self-Identity	Indigeneist Stress-Coping Model	Yes
Knowing Who You Are (Becoming): Effects of a University-Based Elder-Led Cultural Identity Program on Alaska Native Students’ Identity Development, Cultural Strengths, Sense of Community, and Behavioral Health	2023	To provide outcomes of a pilot test of the Cultural Identity Project, an Elder-led cultural identity development program	Psychological Well-Being; Enculturation; Acculturation; Stress; Collectivism	Self-Identified Indigenous Undergraduate Students Who Are Over Eighteen	Alaska Native College Students	University Setting	Multigroup Ethnic Identity Measure–Revise; Native Cultural Health Assessment Tool	Self-Identity Through Recruitment at Indigenous Serving Center	None Specified	Yes
How Ethnic Identity Affects Campus Experience and Academic Outcomes for Native American Undergraduates	2023	To understand how ethnic identity affects campus experience and academic outcomes for Native American Undergraduates	Indigenous identity; Self-Reliance; substance Abuse; Stress	Self-Identification And Current Undergraduate Students Who Are Over Eighteen	Native American Undergraduate Students	University Setting	Multigroup Ethnic Identity Measure (MEIM)	Self-Identity	Social Identity Theory; Tribal Critical Race Theory	No

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
