# Peer review of "Examining Indigenous Identity as a Protective Factor in Mental Well-Being Research in the United States: A Scoping Review"

_ijerph, 2024, doi:10.3390/ijerph21111404_

Round 1
Reviewer 1 Report
Comments and Suggestions for Authors
This paper offers a highly informative and useful contribution to the current literature. The review of existing published research on whether and in what ways Indigenous identity is protective for mental health nicely summarizes and makes sense of mixed results, with good attention to what might drive differences (the tools used in measurement, the life course position of study participants). The methods are clearly described and the geographic focus on the United States is justified. The theoretical framing of Indigenous identity as both political and racialized is well developed, and the author's framework of Tribal Crit is applied in clear and useful ways throughout.
I have only minor suggestions for edits:
- The current formatting of the table is hard to read; I would reformat so that the columns are wider, the spacing is not so wide/uneven between words and the reader can read across more often than "down" the column. This might best be fixed by the journal proofing/formatting team.
- Given the framing that calls attention to Indigenous identity as a political as well as racialized/ethnic identity, I would be interested to see more in the discussion of the measurement tools about which tools incorporate elements of this "dual nature" in the items which they ask, and which ones are more narrowly aligned with (dominant) racialized assessments of identity. This might add a layer to our analysis of how tools shape results. Do tools that have been developed by Indigenous communities/researchers have items that more clearly identify the "political" in the cultural, for example?
- In the conclusion, the authors note a "lack of uniformity in the theories used" by Indigenous identity researchers. This makes it sound like uniformity would be a good thing, a position I would argue against. Based on my reading, I think the authors are trying to make the point that often the theoretical framework is unspecified and/or there is variation in the theories applied; if this is closer to their meaning, I would suggest rewording.
- The word "tenants" in line 70 should be "tenets"
- The authors do a nice job discussing the complexities of including tribal affiliations in data summarizations - there's no one right way to do this. I also appreciate that they look at whether study authors mention their own Indigenous identities. In this context, I would encourage the authors (who clearly identify as Indigenous, given the references to "we") to consider whether they wish to describe the make-up of their own team as Indigenous and/or ally scholars. If the authors wish to include tribal affiliations with their names, I would encourage the journal to allow this - I do not know the journal's policy but am aware that some journals do not accept this as a default.
Author Response
This paper offers a highly informative and useful contribution to the current literature. The review of existing published research on whether and in what ways Indigenous identity is protective for mental health nicely summarizes and makes sense of mixed results, with good attention to what might drive differences (the tools used in measurement, the life course position of study participants). The methods are clearly described and the geographic focus on the United States is justified. The theoretical framing of Indigenous identity as both political and racialized is well developed, and the author's framework of Tribal Crit is applied in clear and useful ways throughout.
I have only minor suggestions for edits:
- The current formatting of the table is hard to read; I would reformat so that the columns are wider, the spacing is not so wide/uneven between words and the reader can read across more often than "down" the column. This might best be fixed by the journal proofing/formatting team.
Response: Thank you for this comment, We have gone back and tried to reformat the table using a horizontal framing. This may help. In addition, We have added a section on the findings of each study, labeled “outcomes”. This will help for people viewing the table.
- Given the framing that calls attention to Indigenous identity as a political as well as racialized/ethnic identity, I would be interested to see more in the discussion of the measurement tools about which tools incorporate elements of this "dual nature" in the items which they ask, and which ones are more narrowly aligned with (dominant) racialized assessments of identity. This might add a layer to our analysis of how tools shape results. Do tools that have been developed by Indigenous communities/researchers have items that more clearly identify the "political" in the cultural, for example?
Response: Thank you for this comment. We have worked to add to the discussion section to embellish on our findings on lines 382-391. When we reviewed the list of identity scales, the large number of ones created for specific Indigenous communities was surprising. That this all happened in the early 2000s was incredible. One challenge with the scales is that they were all developed and utilized before TribalCrit was first published and would not have been able to consider the theory unless the research teams also had come to the same conclusions and implemented them. Second is that outside of these specific studies with the Tribal Nation specific tools, they have never been used again and locating them was rather difficult. To that end, when we evaluated each using TribalCrit, each had its own issues. None properly utilize all aspects of the theory with even those being developed for Tribal Nations having requirements of needing to be enrolled. As a result, we cannot praise certain ones over the others.
- In the conclusion, the authors note a "lack of uniformity in the theories used" by Indigenous identity researchers. This makes it sound like uniformity would be a good thing, a position I would argue against. Based on my reading, I think the authors are trying to make the point that often the theoretical framework is unspecified and/or there is variation in the theories applied; if this is closer to their meaning, I would suggest rewording.
Response: Thank you for catching this. Indeed, we do not wish to encourage a “uniform” set of theories and methods in the field. What we wished to point out is that not all studies are explicit in stating which theories they utilize, whether they are Indigenous theories or not. It is important for transparency as well as showing other researchers and students so they can build upon our work.
- The word "tenants" in line 70 should be "tenets"
Response: Thank you, we have located this and replaced the word.
- The authors do a nice job discussing the complexities of including tribal affiliations in data summarizations - there's no one right way to do this. I also appreciate that they look at whether study authors mention their own Indigenous identities. In this context, I would encourage the authors (who clearly identify as Indigenous, given the references to "we") to consider whether they wish to describe the make-up of their own team as Indigenous and/or ally scholars. If the authors wish to include tribal affiliations with their names, I would encourage the journal to allow this - I do not know the journal's policy but am aware that some journals do not accept this as a default.
Response: Thank you, we will attempt to add our Tribal affiliations to the paper and will consult with the journal to see what the best way to do this is. Our senior authors who are Indigenous have been very consistent in adding Tribal affiliations to their work so we will do the same here.
Reviewer 2 Report
Comments and Suggestions for Authors
Dear Author(s),
This study is interesting due to include current condition of Indigenous populations in both of mental health and their environments as native, political and identity issues. It is distinct that paper contained many information about these points. Therefore, this paper might have two perspective as health and social aspects. However, it seems that the main goal was the potential role of Indigenous identity plays in their mental health. This study focused to examine the effects of Indigenous identity on mental well- being, which is important point as review. So, the study also examines the current literature in health sciences, which focuses on the relationship between United States located Indigenous identity and mental well-being through the lens of TribalCrit (Tribal Critical Race Theory).
This review is limited to only examining measurements of Indigenous identity in the United States. However, the research questions ordinary had been mixed. Once, it is well to probe measuring Indigenous identity as a limitation. After that, the other questions follow to each other as shown:
1. 2.What is known about the relationship between Indigenous identity and mental well-being?
2. 1.What are the measures and instruments researchers are utilizing when measuring Indigenous identity within the U.S.?
3. In which regions of the United States are researchers conducting their work on identity and mental well-being with Indigenous people? As if, such ordinary could be fit for the readers because of defining the Indigenous identity and then probe the relationship between their identity and mental well-being.
Regarding the method, reviewed databases seem proper and elaborate in terms of extensive of the literature to review as in PubMed, CINAHL (EBSCO- host), PsycINFO (EBSCOhost), and ProQuest Dissertations & Theses Global. Similarly, inclusion criteria seem well as published within the United States, English language, full text articles, and United States based Indigenous Peoples, including federally recognized, state recognized, and non-recognized Tribal Nations, as well as Alaska Native and Native Hawaiian. Therefore, this study was limited by studies in U.S. and English language. Also, the figure of the review studies could be shown as in a diagram or an Appendix to show how studies were contained in the present result. Additionally, different data tools used reviewed studies could be the other limitation of this study. Additionally, the paper could be included a content criteria as shown sources types. It was very important that two independent reviewers collected the data non- consulting each other for no bias in terms of the results.
Regarding the Results,
The author(s) found that the impact of a strong Indigenous identity on well-being depended on the questions being examined as well as the age of the populations in terms of the negative, positive or neutral relationships between Indigenous identity and mental well-being. However, the percent of these results should be given in a table to recognize the current results for the readers. This is valid for the Geographical Location and Populations.
Regarding the Discussion,
As a result, this study found that there was negative correlation between Indigenous identity and measurable well-being outcomes as participants improved mental health when they had stronger Indigenous identity in the reviewed studies in particular among the youth populations. However, the discussion of the current result seems weak due not to reference enough previous study findings or expert views. This situation occurs a gap in the paper.
Regarding the implications, in terms of the Indigenous, it could be emphasized for both negative and positive affect identity on the mental health as based the current results. Perhaps, it might be for the generalization of this study findings to focus about the identity concept. For the conclusion, author(s) could be emphasized vital points about the research questions as a total response originated from the current result.
Consequently, I congratulate the author(s) exhibited effort for this study.
Good luck.
Author Response
Dear Author(s),
This study is interesting due to include current condition of Indigenous populations in both of mental health and their environments as native, political and identity issues. It is distinct that paper contained many information about these points. Therefore, this paper might have two perspective as health and social aspects. However, it seems that the main goal was the potential role of Indigenous identity plays in their mental health. This study focused to examine the effects of Indigenous identity on mental well- being, which is important point as review. So, the study also examines the current literature in health sciences, which focuses on the relationship between United States located Indigenous identity and mental well-being through the lens of TribalCrit (Tribal Critical Race Theory).
This review is limited to only examining measurements of Indigenous identity in the United States. However, the research questions ordinary had been mixed. Once, it is well to probe measuring Indigenous identity as a limitation. After that, the other questions follow to each other as shown:
- 2.What is known about the relationship between Indigenous identity and mental well-being?
- 1.What are the measures and instruments researchers are utilizing when measuring Indigenous identity within the U.S.?
- In which regions of the United States are researchers conducting their work on identity and mental well-being with Indigenous people? As if, such ordinary could be fit for the readers because of defining the Indigenous identity and then probe the relationship between their identity and mental well-being.
Response: Thank you for this comment. One of the main goals of this scoping review, as seen in the protocol, is that we were interested to know the location of where this research is taking place. The United States is a very large and geographically diverse country. The Indigenous Peoples within the borders of the United States represent this, in fact, there are 574 federally recognized Tribal Nations, hundreds of Tribal Nations recognized by states, and even more that are not recognized. This means that there are hundreds and hundreds of unique communities and cultures within the umbrella terms of Indigenous Peoples, Native Americans, and American Indians/Alaska Natives. Therefore, while it is interesting to know what the tools and instruments to measure are, we also needed to know with which Tribal Nations and communities are scientists working with. Because we looked for this, we found that all of the identity research in our criteria was done with urban populations in California. This means there is still much that could be done in other large metro areas like Phoenix, Denver, Seattle, New York, and others.
Regarding the method, reviewed databases seem proper and elaborate in terms of extensive of the literature to review as in PubMed, CINAHL (EBSCO- host), PsycINFO (EBSCOhost), and ProQuest Dissertations & Theses Global. Similarly, inclusion criteria seem well as published within the United States, English language, full text articles, and United States based Indigenous Peoples, including federally recognized, state recognized, and non-recognized Tribal Nations, as well as Alaska Native and Native Hawaiian. Therefore, this study was limited by studies in U.S. and English language. Also, the figure of the review studies could be shown as in a diagram or an Appendix to show how studies were contained in the present result. Additionally, different data tools used reviewed studies could be the other limitation of this study. Additionally, the paper could be included a content criteria as shown sources types. It was very important that two independent reviewers collected the data non- consulting each other for no bias in terms of the results.
Response: We are not quite certain what you mean by a limitation here. If you are speaking about the different indexes which articles can be found, in the scoping review process, all of the titles are merged into one large list and then reviewed in bulk, so we are not able to identify which search engine had each specific article. If this about DistillerSR, we utilized a single program, not many, for this. Our research team of 3 reviewers all utilized DistillerSR to conduct their parts of the review process so we had a uniform way to discuss conflicts and discrepancies. The only article I know the location of is the dissertation as that was found in the ProQuest for Dissertations. As far as results being showing in a figure, Figure 1 is the standard way in which reviews document their search strategy and review process flow. We hope this is able to ease your concerns here.
Regarding the Results,
The author(s) found that the impact of a strong Indigenous identity on well-being depended on the questions being examined as well as the age of the populations in terms of the negative, positive or neutral relationships between Indigenous identity and mental well-being. However, the percent of these results should be given in a table to recognize the current results for the readers. This is valid for the Geographical Location and Populations.
Response: Thank you for this suggestion. We have added a new section to the table labeled, “Key Findings”, which will help those reviewing the table. We feel this is more suitable than a simple number or percentage for “positive”, “negative”, “no relationship” as it provides better context which is then embellished in the paper. We hope this addresses your concern.
Regarding the Discussion,
As a result, this study found that there was negative correlation between Indigenous identity and measurable well-being outcomes as participants improved mental health when they had stronger Indigenous identity in the reviewed studies in particular among the youth populations. However, the discussion of the current result seems weak due not to reference enough previous study findings or expert views. This situation occurs a gap in the paper.
Response: This review found that there are equal numbers of papers that show a positive, negative, or no association between Indigenous identity and measures of mental wellbeing. Because we saw equal numbers for each of the three categories, we cannot make a claim in any one direction in our discussion section. This may be due to the lack of instruments that are built by and for Indigenous people in the United States as the history of our populations warrants this.
Regarding the implications, in terms of the Indigenous, it could be emphasized for both negative and positive affect identity on the mental health as based the current results. Perhaps, it might be for the generalization of this study findings to focus about the identity concept. For the conclusion, author(s) could be emphasized vital points about the research questions as a total response originated from the current result.
Response: We cannot imply directionality either way. We can’t give emphasis to either or both of positive and negative impacts of Indigenous identity as we had an entire 1/3 of the articles show no effect of Indigenous identity on mental wellbeing.
Consequently, I congratulate the author(s) exhibited effort for this study.
Good luck.